# *Dendrobium officinale* Polysaccharides Better Regulate the Microbiota of Women Than Men

**DOI:** 10.3390/foods11111641

**Published:** 2022-06-02

**Authors:** Wenyang Tao, Wei Liu, Mingzhe Wang, Wanyi Zhou, Jianrong Xing, Jing Xu, Xionge Pi, Xiaotong Wang, Shengmin Lu, Ying Yang

**Affiliations:** 1Institute of Food Science, Zhejiang Academy of Agricultural Sciences, Hangzhou 310021, China; wenyang_tao@163.com (W.T.); mingzhe_wang1006@163.com (M.W.); munaiyi0612@163.com (W.Z.); 15869132982@163.com (J.X.); lushengmin@hotmail.com (S.L.); 2Institute of Plant Protection and Microbiology, Zhejiang Academy of Agricultural Sciences, Hangzhou 310021, China; biolwei@sina.com (W.L.); pixionge@163.com (X.P.); 3Zhejiang Shouxiangu Pharmaceutical Co., Ltd., Jinhua 321000, China; xujing@sxg1909.com (J.X.); wangxiaotong@sxg1909.com (X.W.)

**Keywords:** *Dendrobium officinale* polysaccharide, genders, intestinal microbiota, short-chain fatty acids (SCFAs), correlation

## Abstract

*Dendrobium officinale* is widely used as a health supplement, but its specific impact on healthy gut microbiota has not yet been clarified, nor has its impact on different human genders. To overcome the problems mentioned above. DOP was extracted and purified with an 8000–12,000 Da dialysis bag. The molecular weight and monosaccharide composition were determined using HPGPC and GC. Gas chromatography was used to detect the content of SCFA. 16S rDNA sequencing was used to analyze the diversity of human microbiota. The results showed that DOP contained two fractions, with an average molecular weight of 277 kDa and 1318 Da, and mainly composed of mannose and glucose. DOP can increase the relative abundance of benign microbiota and decrease the harmful types. Propionic acid content in women was significantly increased after DOP treatment. Finally, the correlation analysis revealed that DOP was beneficial to the microbiota of both men and women. It can be concluded from the results that DOP is a health supplement suitable for humans, and especially women.

## 1. Introduction

*Dendrobium officinale* is distributed in various countries around the world and is widespread in China [1]. *D. officinale* is a traditional medicine of China that is used as a health supplement to prevent various diseases, such as cancer, inflammation, and intestinal barrier damage [2,3,4]. Over a hundred compounds have been isolated from *D. officinale* in recent decades, and polysaccharide is considered the main contributor to its health-promoting activities [5,6]. Several *D. officinale* polysaccharides (DOP) with different structures and configurations have been discovered in recent decades. These polysaccharides are mainly composed of glucose and mannose, with a few galactoses, xyloses, arabinoses, and rhamnoses [7].

Studies have shown that there is a link between intestinal microbes, gender, and immunity, such as a higher risk of type 1 diabetes for women, which is associated with intestinal microbes [8,9]. The composition of the gut microbiome varies between men and women, and the effects on some conditions vary [10]. The microbiota has a close symbiotic relationship with the human host, which plays a vital role in human health, promotes the metabolism of indigestible dietary components, prevents pathogen colonization, and helps the immune system mature [11]. Therefore, confirming the effect in different genders is helpful, to understand the mechanism of functional components more comprehensively.

DOP and other non-starch polysaccharides have a variety of physiological activities, and due to the lack of a variety of polysaccharide hydrolases in the human body, polysaccharides also produce physiological activities through microbiota fermentation and other related pathways [12,13]. With the help of microbiota, SCFA is generated through glycolysis pathways, reducing intestinal pH, and thereby inhibiting harmful bacteria and promoting the growth of beneficial bacteria [14]. Short-chain fatty acids (SCFA) are easily absorbed by the body and play an important role in lowering intestinal pH and inhibiting pathogenic microorganisms [15]. Polysaccharides can alleviate disease symptoms and maintain physiological activity by enhancing the body’s intestinal microbial diversity, regulating the composition of the microbiota, and promoting the growth and proliferation of beneficial bacteria [16]. The production of metabolites is also related to the composition of microbiota. Therefore, the regulatory effect of DOP on the microbiota and SCFA is one of the focuses of polysaccharide activity research. In addition, the differences between men and women remained unclear.

To explore the effect of DOP on the microbiota of different genders, and to provide a reference for the targeted product development of *D. officinale*, the polysaccharide was extracted from the stem, and in vitro fermentation was applied to the polysaccharide, in the present research, to determine the change of short-chain fatty acids after fermentation by HPLC. Moreover, 16S rDNA sequencing was applied to confirm the influence on microbiota relative abundance after fermentation.

## 2. Materials and Methods

### 2.1. Materials and Reagents

Fresh *Dendrobium officinale* is collected from Zhejiang Shouxiangu Pharmaceutical Co., Ltd., Jinhua, China. (N 28°31′, E 119°27′) Zhejiang Province, and the variety was *D. officinale Xianhu 2*. Glucose, ethanol, sulfuric acid, and phenol were all analytically pure and purchased from Sinopharm Chemical Reagent Co., Ltd., Zhejiang, China.

### 2.2. Instruments and Equipment

ICS-5000 Ion Chromatograph was purchased from Diane Corporation, Michigan, USA. DK-8D electric constant temperature sink was purchased from Shanghai Jinghong Experimental Equipment Co., Ltd., Shanghai, China. UV-1800 UV Spectrophotometer was purchased from Shimadzu Instruments Suzhou Co., Ltd., Suzhou, China. The low-speed centrifuge was purchased from Shanghai Anting Scientific Instrument Factory, Shanghai, China. YP2002 Electronic Balance was purchased from Shanghai Hengji Scientific Instrument Co., Ltd., Shanghai, China.

### 2.3. Preparation and Purification of DOP

Fresh dendrobium stems weighing 100 g were added to 1500 mL of water. Then the mix was homogenized and extracted for 2 h under 70 °C, and centrifuge at 4000 r/min for 10 min after extraction to collect the supernatant. The supernatant was added to ethanol according to the volume ratio of 1:4, so that the final volume fraction of ethanol in the solution reached 80% (*v*/*v*); then, stored at 4 °C for 12 h, centrifuged at 3000 r/min for 10 min to separate the residual from the supernatant, and freeze-dried to obtain dendrobium stem crude polysaccharides. The obtained crude polysaccharides were slowly dissolved in water and dialyzed with an 8000–12,000 Da dialysis bag for 72 h, changing the water every 4 h during this period. The dialysate was freeze-dried in vacuum to obtain *Dendrobium officinale* polysaccharides (DOP).

### 2.4. Determination of Molecular Weight of DOP

Molecular weight was determined using high performance gel permeation chromatography (HPGPC). The instrument consisted of a Waters 1525 HPLC system, Waters 2414 refractive index detector, an Empower 3 workstation, and a Waters Ultrahydrogel™ (Waters, Milford, MA, USA) Linear (300 mm × 7.8 mm) column. The mobile phase was NaNO_3_ solution (0.1 mol/L), at a flow rate of 0.9 mL/min and a column temperature of 45 °C.

### 2.5. Monosaccharide Composition of DOP

First, 5 mg of DOP was weighed in acid hydrolyzed vials, then, 1 mL of 2 mol/L methanol hydrochloride solution was added, hydrolyzed at 80 °C for 10 h, and dried with nitrogen. Hydrochloride solution was hydrolyzed at 80 °C for 10 h and dried with nitrogen. Another 1 mL of 2 mol/L trifluoroacetic acid (TFA) was added to the vial for complete acid hydrolysis, and after hydrolysis in an oven at 121 °C for 2 h, the TFA was removed with nitrogen. Next, 300 μL of methenal was added to the residue and removed with nitrogen 3 times. The residual was dissolved with 25 mL diluted water and filtered with a 0.45-μm filter. A CarboPac PA20 and pulse ampere detector were applied for GC analysis. Water, 250 mmol/L NaOH, and 1 mol/L NaAc were selected as mobile phases, at a flow rate of 0.5 min/mL.

### 2.6. Total Polysaccharide Content of DOP

Pipette 1 mL 0.05 mg/mL DOP solution, 1.0 mL distilled water as blank, place in 15 mL plugged test tubes, add 5% phenol solution 0.5 mL, and 5 mL sulfuric acid after sufficient shaking. Shake again and move to 80 °C water bath for 10 min reaction, cool to room temperature. The absorbance was measured at 490 nm, and the sample content was calculated according to the standard curve.

### 2.7. In Vitro Fermentation

Feces were collected from a total of 15 volunteers, 8 men and 7 women, all of whom had no underlying diseases, no long-term consumption of probiotics, and no use of antibiotics and other agents for half a year. Short-term consumption of probiotics or antibiotics within two weeks was also prevented before the experiment. All the volunteers were aged from 20 to 45, and signed an informed consent form. Feces weighing 3 g were collected from each sample. The feces were diluted with PBS to a ratio of 1 g feces: 10 mL PBS (PH = 7.2), mixed well with a shaker, and filtered with gauze, to obtain an original bacterial solution. The original bacterial solutions were used in the following fermentation assay. Precisely 500 μL of 20 mg/mL DOP (95 °C water bath for 1 h to dissolve DOP) was added to 4 mL culture medium, then 500 μL of the original bacterial solution was added, the mixture was mixed well and placed in a 37 °C incubator for 24 h. All the procedures, from collection to the start of fermentation, were performed within 8 h, to ensure the activity of bacteria.

### 2.8. 16S rDNA Sequencing

After extracting genomic DNA from all samples, PCR amplification and product purification were performed, and bacterial 16S rDNA (V3 + V4) was sequenced using the Illumina HiSeq 2500 platform, after quality inspection. Sequencing was done by Majorbio Biotechnology Co., Ltd., Shanghai, China.

### 2.9. Determination of Short Chain Fatty Acid Content

After fermentation was completed, the fermentation broth was centrifuged at 12,000 rpm for 5 min. Aspirate 500 μL of supernatant (precipitate for 16S RNA detection), add to 100 μL of crotonic acid/metaphosphate solution, mix well, and place in a −40 °C refrigerator for 24 h; after acidification, centrifuge the sample at 12,000 rpm for 3 min, filter the supernatant with a 0.22 μm microporous filter membrane, and after filtering, pipette 150 μL into the injection vial for determination of short-chain fatty acids.

### 2.10. Statistical Analysis

SPSS 21.0 was used to statistically analyze the data, and the results were expressed as mean ± standard deviation (mean ± std). The significance difference analysis of microbiota was performed using the Kruskal–Wallis H test or Welch’s *t*-test, with the help of the online platform of Majorbio Cloud Platform (www.majorbio.com, (accessed on 15 April 2020)). Another significant analysis between the groups was performed using an independent sample *t*-test. The significant difference was set to *p* < 0.05.

## 3. Results and Discussion

### 3.1. Preparation and Purification of DOP

With its extraordinary bioactivities such as antitumor activity, anti-inflammatory activity, and immunoactivity, *Dendrobium officinale* has been widely used and consumed as a supplement, to promote better health [17]. The bioactivities are closely related to the structures, especially carbohydrates, which contain different monosaccharides, linkages, and so on [18,19]. Total sugar content was measured using the phenol-sulfuric acid method, with a result of 91.85 ± 2.76%. The remaining 8.15% was considered a result of the different color reaction between the sample and the standard used. The DOP was considered a pure sample. It was shown using HPGFC that *D. officinale* polysaccharides contained two fractions, with an average Mw of 277 KDa and 1318 Da, respectively. The peak-to-area ratio was 75.49:24.51. Too high a molecular weight is not conducive to the physiological activity of polysaccharides. The higher the Mw, the higher entanglement within the molecule, which inhibits the exposure of active groups [20]. A polysaccharide with a molecular weight slightly above 100 kDa is considered of high biological activity [21]. The total polysaccharide content of DOP was determined by the phenol sulfate method, and the total polysaccharide content of DOP was measured as 95%. The monosaccharide composition of the two fractions was determined by ion chromatography. Fucose, arabinose, rhamnose, galactose, glucose, xylose, fructose, mannose, galactoallic acid, glucuronic acid, glucosamine, and glucosamine 11 monosaccharides were used as standard controls. From Table 1, it can be seen that the DOP was composed of five monosaccharides: rhamnose, galactose, glucose, mannose, and galacturonic acid, of which the mannose content was the highest, accounting for 61.10%; glucose was second, accounting for 36.92%; followed by rhamnose, galactose, and galacturonic acid 3. The monosaccharide content of the species was lower, at 0.59%, 0.85%, and 0.56%, respectively. The results are similar to other varieties of *D. Officinale*. For example, the dendrobium polysaccharide extracted by Liang et al. had a molecular weight of 298 kDa, and the ratio of mannose to glucose was 2.2:1 0 [22,23].

### 3.2. The Regulatory Effect of DOP on the Microbiota

There are tens of thousands of microorganisms in the human intestine, which play an important role in host nutrient metabolism, drug metabolism, maintaining the integrity of the intestinal mucosa, immunity regulation, and resistance to pathogenic bacteria.

Thirty samples from four groups were analyzed using 16S rDNA sequencing. After quality-filtering with QIIME, 1,543,203 clean sequences were generated by 16S rDNA sequencing and the highest 341 OTUs were obtained. To better describe the influence of DOP on microbiota, the α-diversity, including Shannon, Chao1, and Simpson indexes, was analyzed, and is presented in Figure 1. A higher Shannon index represents higher richness, while a higher Chao1 index represents higher evenness. Simpson index is applied to evaluate both richness and evenness at the same time. The Simpson index of the female microbiota treated with DOP was significantly lower than the control. No significant differences were found in the other groups. DOP had a greater effect on the microbiota in women than in men.

It can be seen from the PCA analysis in Figure 2 of the phylum level and the genus level that, whether DOP was added or not, few differences were found in the distribution of men on the PCA map. With more than half of the overlap of both, DOP did not have a significant effect on the PCA result for the male microbiota. The effect of DOP on the women’s group was more pronounced than that of the men, and the overlap between the DOP group and the blank group was less than 50% at the phylum level. This is, again, consistent with previous results.

The relative abundance of microbiota at phylum and genus levels are presented in Figure 3. The feces mainly include Actinobacteria, Bacteroidetes, Firmicutes, and Proteobacteria, which are considered common microbiota [11]. The relative abundance of Bacteroides increased significantly, the ratio of Firmicutes to Bacteroides decreased significantly, and the relative abundance of Proteobacteria decreased significantly in both men and women. The ratio of Firmicutes to Bacteroidetes is associated with obesity [24], and DOP lowers the ratio of Firmicutes to Bacteroidetes, which might help alleviate the symptoms of obesity. Bacteroides were increased under the influence of DOP in both men and women. Similar results were provided by Zhang et al. [25], when they introduced DOP to perimenopausal mice. Some species of Proteobacteria produce endotoxins that can cause discomfort in humans [26]. DOP significantly reduces the relative abundance of Proteobacteria and may have the function of relieving the discomfort caused by endotoxins, which is worthy of further investigation. Actinobacteria are considered beneficial for regulating serum cholesterol levels, modulating the immune system, and inhibiting intestinal diseases [27]. As shown in Figure 3, the relative abundance of Actinobacteria was shifted by the DOP treatment, indicating a potential health-promoting activity of Actinobacteria. The effect of DOP on the relative abundance of the phylum level in women was larger than in men. Women might make better use of DOP, which is consistent with the α-diversity analysis.

The relative abundance at the genus level that was higher than 0.01 is shown in Figure 4. To better describe the changes at the genus level, a heatmap was also created and is presented in Figure 4. As can be seen from Figure 4, the relative abundance of some specific genera was significantly changed, both in men and women. *Enterococcus* significantly increased after the DOP treatment, while *Lachnospiraceae UCG-004* and *Lachnoclostridium* significantly decreased. *Enterococcus* was reported to produce bacteriocins, which are now being considered as a probiotic and a promising alternative for fighting emerging antimicrobial resistance [28]. Therefore, DOP might enhance the ability to combat antimicrobial resistance. *Lachnoclostridium* correlated positively with FE1, which is an indicator of chronic pancreatitis [29], showing the potential of DOP to attenuate chronic pancreatitis.

*Bifidobacterium*, *Bacteroides*, and *Lactobacillus* were increased after DOP treatment, in both men and women. *Lactobacillus* and *Bifidobacterium* were shown to improve physiological function and cognitive ability in aged mice [30]. *Bacteroides* can improve different intestinal disorders, as well as cardiovascular disease, behavior disorders, and cancer, and have gained authorization for food processing by the European Commission, after safety evaluation [31]. These three genera are all considered probiotics, which means DOP is beneficial to human health, regardless of gender. Moreover, acetic acid is the main end-product produced by *Bifidobacterium* and *Lactobacillus*, which might be the reason for the increment of acetic acid in SCFA [32].

Furthermore, some genera acted differently between men and women. *Streptococcus* increased in women, while in men it did not. *Streptococcus thermophilus*, which belongs to *Streptococcus*, is considered a probiotic, to generate an antioxidant and antibacterial activity [33]. *Prevotella 9* increased in men under the treatment of DOP, while in women it did not. Higher relative abundance of *Prevotella* reflects a higher carbohydrate metabolism [34].

### 3.3. Effect of DOP on the Production of Short-Chain Fatty Acids

Short-chain fatty acids (SCFA) are a class of saturated fatty acids containing 1–6 carbon backbones, which can be produced by fermentation of plant carbohydrates under the action of microbiota, and which have been studied in a large number of literature works [35]. The production of SCFA regulates the pH in the human intestine, which in turn regulates the proportion of beneficial and harmful bacteria in the intestine and promotes the absorption of other nutrients [36]. In addition, the bioactivities of SCFA, such as providing energy, exerting antibacterial activity, and maintaining the gut barrier have been widely investigated [37,38]. As can be seen in Figure 5, no significant effect was found on all six different SCFA and total SCFA content, when the men’s and women’s groups were combined. After analyzing the two genders separately, DOP could significantly promote the production of acetic acid and propionic acid in women, and eventually lead to a significant increase in the total short-chain fatty acid content. Fu et al. also applied DOP to carry out in vitro fermentation experiments and found that DOP can promote the production of acetic acid, propionic acid, and butyric acid [39]; the more pronounced differences in the results may be related to the longer fermentation time. Acetic acid stimulates the intestinal secretion of intestinal hormones, such as glucagon-like peptide-1 (GLP-1) and peptide YY, to promote host energy and substrate metabolism, by reducing systemic lipolysis and increasing energy expenditure, promoting fat oxidation, and reducing the level of systemic pro-inflammatory cytokines, to affect appetite [40]. DOP may have the effect of regulating appetite according to this theory. Data confirmed that propionic acid can induce apoptosis of cancer cells in vitro [41]. Whether DOP has the effect of inducing apoptosis in cancer cells deserves further study. Li et al. provided a hypothesis that DOP provides immunomodulatory activities, by significantly improving the production of butyrate; while butyrate was increased in women, without significance, in the present research [1]. The conflicting results might be ascribed to the different models and the dosages used in the two studies.

### 3.4. Correlation between SCFA, DOP, and Microbiota

To better evaluate the relationships between SCFA and DOP, an RDA analysis was applied to the different genders. It can be seen in Figure 6A that the vector center was moved from the 4th quadrant to the 2nd quadrant after DOP treatment in men. Moreover, interestingly, the vector center without DOP treatment was negatively related to all SCFA, while the vector center with DOP treatment was positively related to all SCFA. The vector center of women was moved from the 1st quadrant to the 3rd quadrant after DOP treatment (Figure 6B). In addition, similarly for men, the vector center without DOP treatment was negatively related to all SCFA, while the vector center with DOP treatment was positively related to all SCFA. These data showed that DOP, within the scope of RDA analysis, promoted all samples to produce SCFA. Combined with the results of SCFA content, DOP could improve the SCFA content of all samples and have a greater effect in women.

The correlation between microbiota and SCFA was analyzed and is presented as a Spearman correlation heatmap in Figure 6C. Several bacteria were strongly related to SCFA. *Bacteroides* were significantly positively related to propionic acid. *Prevotella 2* was significantly positively related to isobutyric acid. *Lactobacillus* was significantly positively related to isovaleric acid. *Desulfovibrio* and *Megasphaera* were significantly positively related to valeric acid. *Escherichia-Shigella* and *Paraclostridium*, which were decreased under the treatment with DOP, were significantly negatively related to acetic acid and propionic acid. The results were also consistent with the previous results of SCFA and the relative abundance of microbiota. *Escherichia-Shigella* was also significantly negatively related to total SCFA content, which might be attributed to its relationship with propionic acid and its high relative abundance in microbiota.

## 4. Conclusions

*Dendrobium officinale* polysaccharide was extracted and purified in the present research. The impact of DOP on microbiota from different genders was analyzed with 16S rDNA sequencing. Changes in SCFA contents were compared between the genders. Finally, the correlation between SCFA, DOP, and microbiota was analyzed. The results showed that DOP contained two fractions, with an average Mw of 277 KDa and 1318 Da, and mainly consisting of mannose and glucose. DOP has a greater effect on the microbiota of women than men, from the scope of α, β diversity and the relative abundance of microbiota. DOP significantly promoted the production of propionic acid in women. In a word, DOP is a health supplement suitable for humans, particularly for women.

## Figures and Tables

**Figure 1 foods-11-01641-f001:**
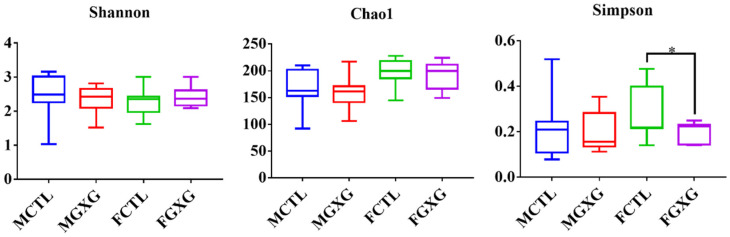
Effect of *Dendrobium officinale* polysaccharides on the α-diversity of microbiota. * means *p* < 0.05.

**Figure 2 foods-11-01641-f002:**
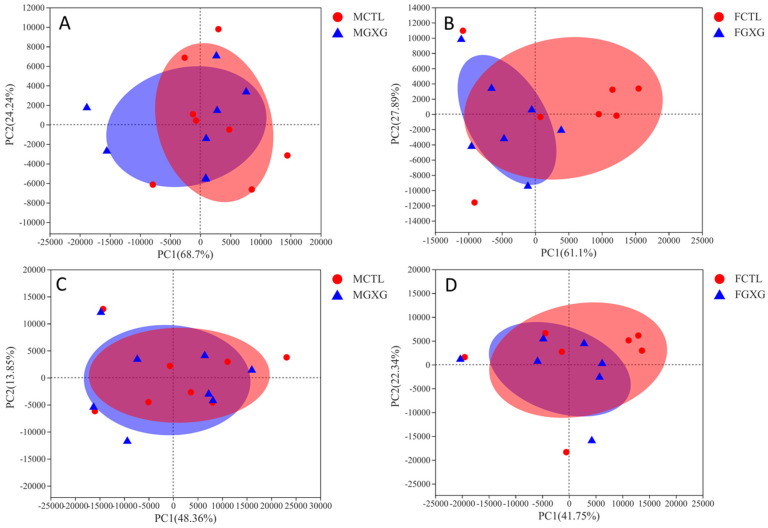
PCA analysis of the effect of *Dendrobium officinale* polysaccharides on microbiota composition. (**A**) men’s phylum level; (**B**) women’s phylum level; (**C**) men’s genus level; (**D**) women’s genus level.

**Figure 3 foods-11-01641-f003:**
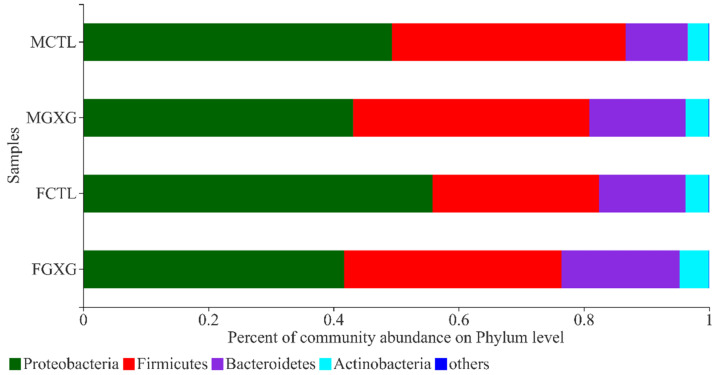
Effect of *Dendrobium officinale* polysaccharides on the relative abundance of microbiota at the phylum level.

**Figure 4 foods-11-01641-f004:**
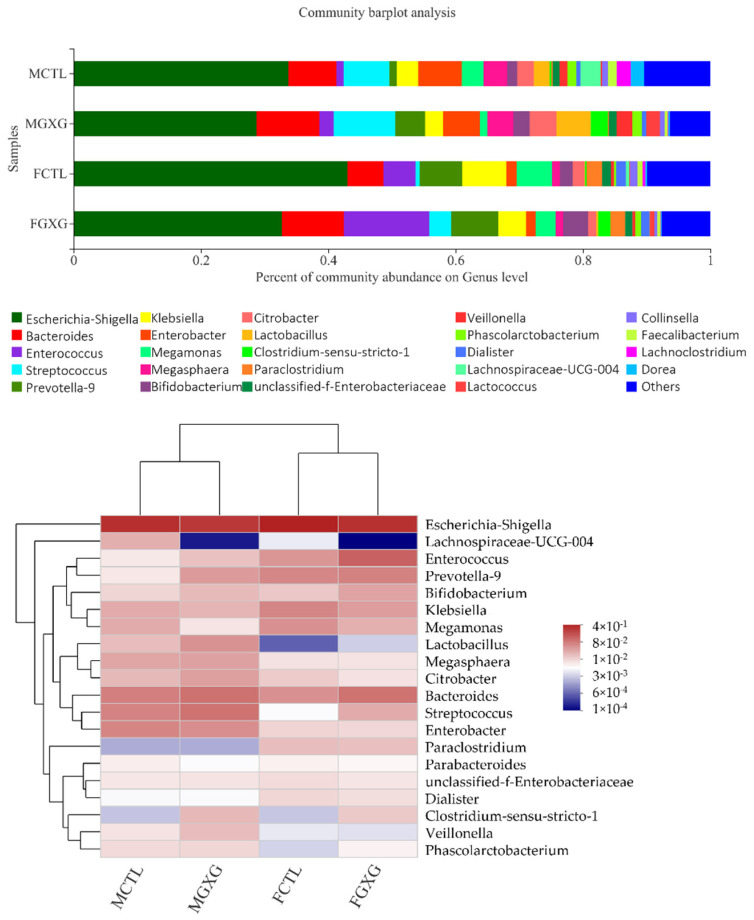
The relative abundance (**up**) and heatmap (**down**) of microbiota, with/without *Dendrobium officinale* polysaccharide treatment, at the genus level.

**Figure 5 foods-11-01641-f005:**
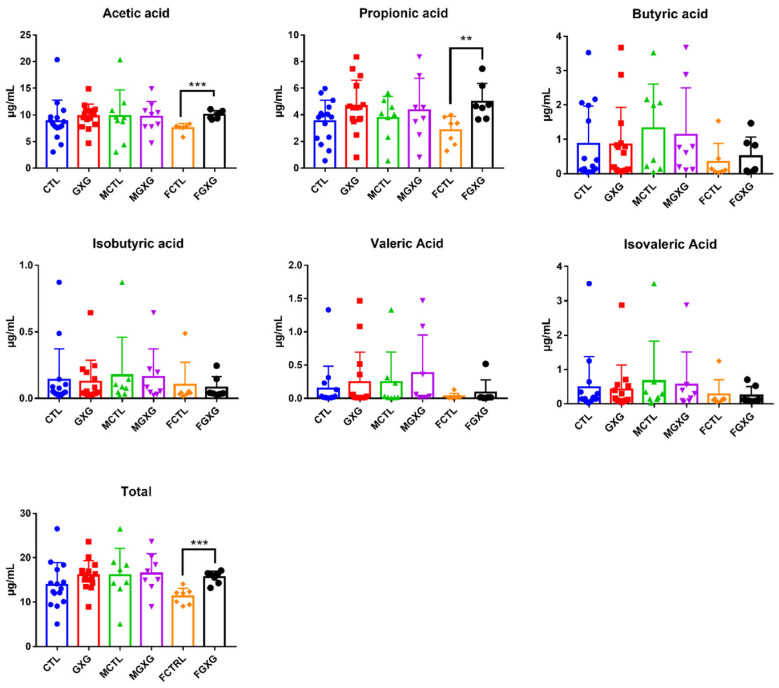
Effect of *Dendrobium officinale* polysaccharides on the production of short-chain fatty acids in the microbiota. ** *p* < 0.01. *** *p <* 0.005.

**Figure 6 foods-11-01641-f006:**
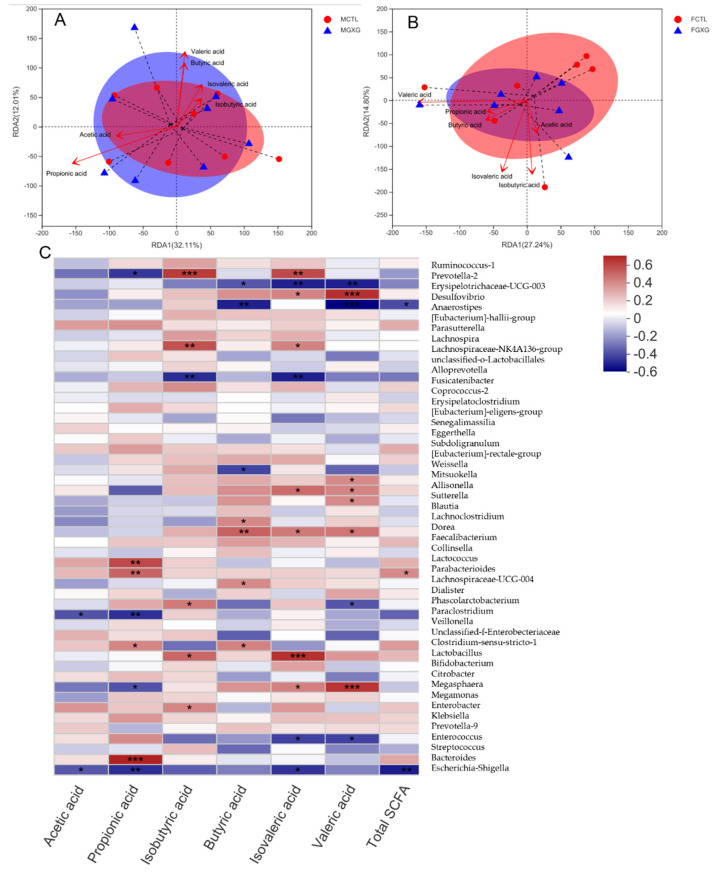
RDA analysis between SCFA and DOP of (**A**) men and (**B**) women. (**C**) Correlation between SCFA and microbiota. * *p* < 0.05. ** *p* < 0.01. *** *p* < 0.005.

**Table 1 foods-11-01641-t001:** The monosaccharide composition of Dendrobium polysaccharide.

Peak	Name	Retention Time (min)	Peak Area (nC×min)	mor%
1	Rhamnose	7.35	0.30	0.59
2	Galactose	9.20	0.69	0.85
3	Glucose	10.54	23.33	36.92
4	Mannose	12.80	41.54	61.10
5	Galacturonic acid	21.61	0.19	0.56

## Data Availability

Data is contained within the article.

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
