# Peer review of "Dendrobium officinale Polysaccharides Better Regulate the Microbiota of Women Than Men"

_foods, 2022, doi:10.3390/foods11111641_

Round 1

Reviewer 1 Report

To explore the effect of DOP on the microbiota of different genders, and to provide a reference for the targeted product development of Dendrobium officinale, the polysaccharide was extracted from the stem, and in vitro fermentation is applied to the polysaccharide in the present research to determine the change of short-chain fatty acids after fermentation by HPLC. Also, 16s rDNA sequencing is applied to confirm the influence on microbiota relative abundance after fermentation.

Introduction is described very well.

Material and methods:Line 64 and other parts of Dendrobium officinale must be in italic.

Producers of chemical, equipment must have a state and place of firm.

15 volunteers, 8 males and 7 females, need more detailed information.

Bacterial isolation needs more description, how were samples prepared etc.

Results are described very well, but need correction of latin names.

No discussion and conclusion is described.

Author Response

Thanks for the reviewer’s kindly advices. The manuscript was revised as comments suggested. All the changes were done using ‘Track Change’ function of MS word.

Reviewer #1

To explore the effect of DOP on the microbiota of different genders, and to provide a reference for the targeted product development of Dendrobium officinale, the polysaccharide was extracted from the stem, and in vitro fermentation is applied to the polysaccharide in the present research to determine the change of short-chain fatty acids after fermentation by HPLC. Also, 16s rDNA sequencing is applied to confirm the influence on microbiota relative abundance after fermentation.

1.1 Introduction is described very well.

Response: Thank you for your compliment.

1.2 Material and methods: Line 64 and other parts of Dendrobium officinale must be in italic.

Response: Thank you for your kindly comment. All the phrase ‘Dendrobium officinale’ in the revised manuscript are in italic now.

1.3 Producers of chemical, equipment must have a state and place of firm.

Response: Thanks for the comment. The state or place of producers were added in the revised manuscript.

1.4 15 volunteers, 8 males and 7 females, need more detailed information.

Response: Thanks for the comment. Feces were collected from a total of 15 volunteers, 8 males and 7 females, all of whom had no underlying diseases, no long-term consumption of probiotics, and no use of antibiotics and other agents for half a year. Short-term consumption of probiotics or antibiotic within two weeks were also prevented before the experiment. All the volunteers were aged from 20-45, signed an informed consent form. The detailed information was added in the revised manuscript.

1.5 Bacterial isolation needs more description, how were samples prepared etc.

Response: Thanks for the comment. The feces weighed 3 g were collected from each sample. The feces were diluted with PBS according to the ratio of 1 g feces: 10 mL PBS (PH=7.2), mixed well with a shaker, and filtered with gauze to obtain the original bacterial solution. The original bacterial solutions were used in the following fermentation assay. All the procedure from collection to the start of fermentation was performed within 8 h to ensure the activity of bacterial. The detailed procedure was added in the revised manuscript.

1.6 Results are described very well, but need correction of latin names.

Response: Thank you for your compliment. The latin names were correctly rephrased in Italic form throughout the revised manuscript.

1.7 No discussion and conclusion is described.

Response: Thank you for your comment. The discussions were added in 3. Results and discussions, and the conclutions were added in 4. Conclusion. The detailed revisions were presented in the revised manuscript.

Reviewer 2 Report

Dear Authors,

This manuscript is devoted to polysaccharides isolated from Dendrobium officinale and their effect on the regulation of short-chain fatty acids production and gut microbiota of different genders. The biological effects of these polysaccharides seems to be very interesting, but the characterisation of isolated product in chemical point of view is insufficient. According to cited review (ref. No 7), there are many Dendrobium polysaccharides of different structure and composition. The polysaccharide fraction reported in the manuscript contained mannose and glucose (about 2:1) mainly, and the other sugars were rather minor components. Therefore, the obtained product could be assigned as glucomannan and compared with other plant glucomanans like konjac gum. However, more analyses should be performed to obtained structural information about the structure of this polysaccharide, i.e. configuration of glycosidic bonds and unit distribution/branching. Methylation analysis and correlation NMR spectra are highly recommended for obtaining of more structural information. In addition, the English grammar and style should be carefully corrected, and the manuscript style should be in accordance with the rules of Foods.

In addition, there are some minor points for revision:

  1. The latinic name of plant species should be in should be written in italics.
  2. The HPGFC chromatogram should be represented to illustrate molecular mass distribution. These are two Mw fractions but not polysaccharides! The polysaccharides could be structurally similar.
  3. What about the purity of this product? How many total sugars, proteins and other components have been found?
  4. Table 1: the last three raws are not informative and should be deleted; the units of Retention time should be added. The relative amounts of monosaccharide units should be represented as mol % or mass %.
  5. The last section is rather Conclusions than Discussion, in should be renamed. The last sentences about this section (lines 305-307, from template) should be deleted.

Author Response

Thanks for the reviewer’s kindly advices. The manuscript was revised as comments suggested. All the changes were done using ‘Track Change’ function of MS word.

Reviewer 2

Dear Authors,

This manuscript is devoted to polysaccharides isolated from Dendrobium officinale and their effect on the regulation of short-chain fatty acids production and gut microbiota of different genders. The biological effects of these polysaccharides seems to be very interesting, but the characterisation of isolated product in chemical point of view is insufficient. According to cited review (ref. No 7), there are many Dendrobium polysaccharides of different structure and composition. The polysaccharide fraction reported in the manuscript contained mannose and glucose (about 2:1) mainly, and the other sugars were rather minor components. Therefore, the obtained product could be assigned as glucomannan and compared with other plant glucomanans like konjac gum. However, more analyses should be performed to obtained structural information about the structure of this polysaccharide, i.e. configuration of glycosidic bonds and unit distribution/branching. Methylation analysis and correlation NMR spectra are highly recommended for obtaining of more structural information. In addition, the English grammar and style should be carefully corrected, and the manuscript style should be in accordance with the rules of Foods.

Response: Thanks for the comments. We’ve aware of the insufficient in structure characterization. The detailed analysis, including methylation analysis, NMR, FT-IR and et al, was on the schedule. We are planning to carry out the procedure after a further separation of the two fractions. The English grammar and style were revised to better express ourselves in the re submitted manuscript.

In addition, there are some minor points for revision:

2.1 The latinic name of plant species should be in should be written in italics.

Response: Thank you for your comment. All the latinic names were in property form now in the revised manuscript.

2.2 The HPGFC chromatogram should be represented to illustrate molecular mass distribution. These are two Mw fractions but not polysaccharides! The polysaccharides could be structurally similar.

Response: Thank you for your comment. The expressions about two Mw fractions were corrected throughout the manuscript.

2.3 What about the purity of this product? How many total sugars, proteins and other components have been found?

Response: Thanks for your comment. Protein content was measured with Bradford assay and no protein was revealed in the samples. Total sugar was measured by phenol-sulfuric acid method with a result of 95.64%. The remaining 4.36% is considers as a result of the slightly different color reaction between samples and the standard we used. The sample was considered pure polysaccharide for these reasons. The information was added in the revised manuscript.

2.4 Table 1: the last three raws are not informative and should be deleted; the units of Retention time should be added. The relative amounts of monosaccharide units should be represented as mol % or mass %.

Response: Thanks for your comment. Table 1 was revised as suggested and the units were added. The relative amounts were expressed as mol% now in the revised manuscript.

2.5 The last section is rather Conclusions than Discussion, in should be renamed. The last sentences about this section (lines 305-307, from template) should be deleted.

Response: Thanks for your comment. The last section is now in the correct form that ends in conclusion.

Reviewer 3 Report

tThe manuscript is interesting, however is in a state of almost unreadability on the results, wich to be more organized and results analysis separated from the results discussion and literature discussion. There is some specific comments in PDF annexed.

Needs to be more developed (results and discussion) to be readable and scientific robust

Author Response

Thanks for the reviewer’s kindly advices. The manuscript was revised as comments suggested. All the changes were done using ‘Track Change’ function of MS word.

The manuscript is interesting, however is in a state of almost unreadability on the results, wich to be more organized and results analysis separated from the results discussion and literature discussion. There is some specific comments in PDF annexed.

Needs to be more developed (results and discussion) to be readable and scientific robust

Response: Thanks for your comments. The comments mentioned in PDF were settled in the revised manuscript.

Round 2

Reviewer 1 Report

Th authors accpeted all sugestion and comments.

Reviewer 2 Report

The revised manuscript is now suitable for publication in Foods.

Reviewer 3 Report

The manuscript was improved although there is some consideration in PDF that should be addressed by the authors.
